# Potential Utility of Pre-Emptive Germline Pharmacogenetics in Breast Cancer

**DOI:** 10.3390/cancers13061219

**Published:** 2021-03-11

**Authors:** Philip S. Bernard, Whitney Wooderchak-Donahue, Mei Wei, Steven M. Bray, Kevin C. Wood, Baiju Parikh, Gwendolyn A. McMillin

**Affiliations:** 1ARUP Institute for Clinical and Experimental Pathology, Salt Lake City, UT 84108, USA; phil.bernard@hci.utah.edu (P.S.B.); whitney.l.donahue@aruplab.com (W.W.-D.); 2Department of Pathology, University of Utah, Salt Lake City, UT 84112, USA; 3Huntsman Cancer Institute, University of Utah, Salt Lake City, UT 84112, USA; mei.wei@hci.utah.edu; 4Division of Oncology, Department of Internal Medicine, University of Utah, Salt Lake City, UT 84112, USA; 5LifeOmic Inc., Indianapolis, IN 46202, USA; steven.bray@lifeomic.com (S.M.B.); kevinwood16@gmail.com (K.C.W.); b.parikh8@gmail.com (B.P.)

**Keywords:** breast cancer, CYP2D6, genotyping, pharmacogenomics, supportive care

## Abstract

**Simple Summary:**

Breast cancer outcomes are variable due to differences in tumor biology, patient biology, and treatment. The likelihood of developing cancer and other diseases increases with age. Thus, many patients with breast cancer have multiple co-morbidities requiring medical management, which increases the probability of polypharmacy and the risk of adverse drug events. Pharmacogenetics is the study of how inherited genetic variants influence drug response. Depending on the genes that a patient inherits, some respond to drugs as expected, some experience debilitating side effects, and others have minimal to no response. In this paper, we discuss the theoretical clinical utility of pharmacogenetics for 225 patients with breast cancer relative to anti-cancer drugs and non-cancer drugs. For this population, 38 drug–gene associations with high levels of evidence for clinical actionability were identified, supporting the concept of pharmacogenetics integration into the routine care of future patients with breast cancer.

**Abstract:**

Patients with breast cancer often receive many drugs to manage the cancer, side effects associated with cancer treatment, and co-morbidities (i.e., polypharmacy). Drug–drug and drug–gene interactions contribute to the risk of adverse events (AEs), which could lead to non-adherence and reduced efficacy. Here we investigated several well-characterized inherited (germline) pharmacogenetic (PGx) targets in 225 patients with breast cancer. All relevant clinical, pharmaceutical, and PGx diplotype data were aggregated into a single unifying informatics platform to enable an exploratory analysis of the cohort and to evaluate pharmacy ordering patterns. Of the drugs recorded, there were 38 for which high levels of evidence for clinical actionability with PGx was available from the US FDA and/or the Clinical Pharmacogenetics Implementation Consortium (CPIC). These data were associated with 10 pharmacogenes: *DPYD, CYP2C9, CYP2C19, CYP2D6, CYP3A5, CYP4F2, G6PD, MT-RNR1, SLCO1B1,* and *VKORC1*. All patients were taking at least one of the 38 drugs and had inherited at least one actionable PGx variant that would have informed prescribing decisions if this information had been available pre-emptively. The non-cancer drugs with PGx implications that were common (prescribed to at least one-third of patients) included anti-depressants, anti-infectives, non-steroidal anti-inflammatory drugs, opioids, and proton pump inhibitors. Based on these results, we conclude that pre-emptive PGx testing may benefit patients with breast cancer by informing drug and dose selection to maximize efficacy and minimize AEs.

## 1. Introduction

Breast cancer outcomes are variable due to differences in tumor biology [1,2], patient biology [3,4], and treatment availability [5,6]. In addition to anti-cancer treatment, cancer patients often receive medications for co-morbidities and supportive care [7], resulting in polypharmacy and an increased likelihood of adverse events (AEs), therapeutic failure, non-standard dose requirements, and/or non-adherence [8,9]. Most cancer patients take five or more prescription and non-prescription drugs at one time [10]. Some combinations of drugs have potential to interact with one another (drug–drug interactions), which can contribute to AEs [11]. In addition, inherited (germline) variations in the genes involved in drug response may independently impact drug response (drug–gene interactions) and/or may exacerbate drug–drug interactions [12]. 

“Pharmacogenetics” (PGx) and “pharmacogenomics” refer to germline variants in genes that are involved in drug response. PGx generally describes single gene–drug associations, whereas pharmacogenomics considers the impact of many genes, as in the genome. The use of PGx can inform prescribing decisions through predicting discrete aspects of pharmacokinetics and/or pharmacodynamics, the two major processes responsible for drug response and AEs [9,13,14]. Pharmacokinetics describes how the body absorbs, distributes, metabolizes, and eliminates a drug, whereas pharmacodynamics describes the physiology responsible for both the desirable and undesirable effects of a drug. A well-known PGx target related to pharmacokinetics is the cytochrome P450 (CYP) drug-metabolizing enzyme family. Each enzyme is associated with either the activation or inactivation of specific drugs. The most common CYP isozymes associated with drug metabolism are CYP2C9, CYP2C19, CYP2D6, CYP3A4, and CYP3A5, which are coded from genes with the same names (*CYP2C9, CYP2C19, CYP2D6, CYP3A4,* and *CYP3A5*, respectively). A person that inherits PGx variants associated with extremes of metabolism may require non-standard dosing or may be best served by avoiding certain drugs [15]. 

The Clinical Pharmacogenetics Implementation Consortium (CPIC) has published levels of evidence for the clinical actionability of PGx drug–gene pairs, the highest of which are levels A and B (https://cpicpgx.org/genes-drugs/) (accessed on 1 March 2021). Assignment of level A or B indicates that prescribing action such as alternative therapies or non-standard dosing is recommended. CPIC Level C and D PGx associations are not associated with any prescribing changes based on genetics. In addition, the US FDA has recently published a Table of Pharmacogenetic Associations organized into three tiers of relevance, with Tier 1 being most clinically actionable (https://www.fda.gov/medical-devices/precision-medicine/table-pharmacogenetic-associations) (accessed on 1 March 2021). Looking at just the CPIC level A and B associations and the content of the three FDA tiers, there are 198 drug–gene pairs, of which 50 overlap (Appendix A). Of these 198 associations, nearly half (93) include CYP genes. Of these 93 CYP associations, 87 involve *CYP2C9*, *CYP2C19,* and/or *CYP2D6*. One reason for the extensive attention on these genes is their widespread involvement in drug metabolism, but also their relatively high prevalence of genetic variation that impacts drug metabolism. CYP gene variants lead to five standardized categories of predicted metabolic phenotypes, ranging from no enzyme activity to very high enzyme activity: poor metabolizer (PM), intermediate metabolizer (IM), normal metabolizer (NM), rapid metabolizer (RM), and ultra-rapid metabolizer (UM) [16,17]. The 198 drug–gene pairs cited by the CPIC also include genes that code for other drug-metabolizing enzymes (e.g., *DPYD, TPMT, UGT1A1*); genes that code for proteins involved in other aspects of drug response such as drug transporters (e.g., *SLCO1B1*); and genes that code for pharmacodynamics effectors, such as receptors (e.g., *SCN1A, RYR1*) and immune mediators (e.g., *HLA-A, HLA-B*).

In this retrospective study, the potential for drug–gene interactions in patients with breast cancer was investigated in relation to both anti-cancer and non-cancer therapy. Specifically, the prevalence and significance of germline variants associated with the 198 drug–gene associations relevant to a cohort of 225 patients with breast cancer were discussed. Although the PGx data were not available to influence patient care decisions in real time, we propose that the inclusion of PGx information in the medical management of patients with cancer, particularly those patients receiving polypharmacy, could have informed and potentially improved prescribing decisions to reduce drug-related AEs and to achieve optimal treatment response.

## 2. Results

### 2.1. Subjects in Breast Cancer Cohort

This retrospective study of 225 breast cancer patients treated at the Huntsman Comprehensive Cancer Center (Salt Lake City, UT, USA) consisted of 53 patients with Stage I, 105 patients with Stage II, 48 patients with Stage III, and 19 patients with Stage IV disease. This patient cohort predominantly identified as white/Caucasian (205/225, 91%). The median age of diagnosis was 51. The median follow-up for the cohort was 49 months. One-hundred-and-sixty-seven patients (74%) had ER-positive breast cancer, and 148 patients (66%) had received adjuvant endocrine therapy (e.g., tamoxifen or aromatase inhibitors). Fifty-three patients (24%) had HER2-positive breast cancer, and 48 patients (21%) had received HER2 monoclonal antibody (e.g., trastuzumab). Two-hundred-and-twenty-one (98%) patients had received chemotherapy. The clinicopathologic characteristics of the patients are provided in Appendix A. 

### 2.2. Pharmacy Trend Analysis

Drug–gene associations with level A or B CPIC evidence for clinical actionability and drug–gene associations included in one or more of the FDA’s Table of Pharmacogenetic Associations were compiled (Appendix A) and then interrogated across the cohort in LifeOmic’s Precision Health Cloud. Of the 198 drug–gene associations identified with high levels of evidence, there were 38 drugs prescribed to patients in this study cohort. There were three anti-cancer drugs that were correlated with two pharmacogenes (Table 1) and 35 non-cancer drugs that were correlated with nine pharmacogenes (Table 2). Of the 38 drugs, 27 (71%) were associated with *CYP2C9*, *CYP2C19*, and/or *CYP2D6*. The remaining pharmacogenes associated with these 38 drugs included other drug-metabolizing enzymes (*CYP3A5, DPYD, G6PD)* and additional genes associated with drug response (*CYP4F2, MT-RNR1, SLCO1B1* and *VKORC1*). The number of patients in the cohort that were documented to have received each drug is included in Table 1 and Table 2. Common indications for the non-cancer drugs were related to a wide range of medical specialties such as psychiatry, pain management, cardiology, infectious disease, and gastroenterology.

### 2.3. PGx Genotyping Panel and CYP2D6 Copy Number Variation Analysis

Of the 225 samples tested, nearly all samples performed well analytically using both the genotyping and *CYP2D6* copy number assays for 8 of the 10 pharmacogenes included in Table 1 and Table 2. The performed PGx testing was not designed to interrogate *G6PD* or *MT-RNR1.* Data were summarized for the PGx results when all acceptance criteria for the analytical data were met, and phenotypes were predicted based on consensus nomenclature. Reasons for unacceptable results included poor quality and/or quantity of DNA, inadequate amplification, or inconclusive data. Actionable variants were detected in at least one of the eight pharmacogenes tested in this study for all patients. The patients that were noted to be using each drug were further evaluated to identify the percentage of patients with abnormal phenotype predictions, such as altered metabolism or non-standard response regarding anti-cancer drugs (Table 1) and non-cancer drugs (Table 2). 

For the anti-cancer drugs, no PGx variants were identified in *DYPD*, which is relevant to 5-fluorouracil and capecitabine (CPIC Level A, FDA Tier 1), but *CYP2D6* variants associated with clinically relevant phenotypes were predicted for approximately half of those patients (51%) prescribed tamoxifen (CPIC Level A, FDA Tier 3). 

Non-cancer drug classes that were prescribed to at least one-third of patients (56/225) included anti-depressants, anti-infectives, non-steroidal anti-inflammatory drugs (NSAIDs), opioids, and proton pump inhibitors (Table 2). The most commonly prescribed drugs included NSAIDs (ibuprofen (CPIC Level A, *n* = 144) and celecoxib (CPIC Level A, FDA Tier 1, *n* = 116)), opioids (hydrocodone (CPIC Level B, *n* = 127) and tramadol (CPIC Level A, FDA Tier 1, *n* = 99)), and the proton pump inhibitor omeprazole (CPIC Level A, FDA Tier 3, *n* = 83). Variant PGx phenotypes were common for *CYP2C9* (NSAIDs), *CYP2C19* (omeprazole) and *CYP2D6* (opioids). Using omeprazole as an example, the PGx data predicted that 48% of patients prescribed omeprazole would exhibit non-standard metabolism. Because omeprazole is inactivated by CYP2C19, reduced metabolism is associated with adverse events, whereas rapid metabolism is associated with poor efficacy. Of the 83 patients prescribed omeprazole, there were 23 RMs and 2 UMs that might have benefited from increased dosing of omeprazole. Likewise, there were 14 IMs and 1 PM that might have benefited from decreased dosing of omeprazole. No changes to clinical practice could be made for this cohort based on PGx associations because the data for the study were all collected retrospectively. 

The overall prevalence of *CYP2C9*, *CYP2C19*, and *CYP2D6* phenotypes is shown in Table 3. These data demonstrate a relatively high frequency of abnormal metabolic phenotypes among this patient cohort, consistent with published allele frequencies for Caucasians. As such, patients requiring polypharmacy could theoretically benefit from pre-emptive PGx testing. Variants in the genes that code for glucose 6-phosphate dehydrogenase (*G6PD*) and for mitochondrial 12S rRNA (*MT-RNR1*) are rare in Caucasian populations such as this cohort, although they could potentially be clinically important in other populations [18,19,20].

## 3. Discussion

The purpose of this study was to demonstrate the potential relevance of pre-emptive PGx testing in a cohort of 225 patients with breast cancer. Polypharmacy is common amongst this cohort and many patients have PGx associations with high levels of evidence for actionability. Three patient case examples were selected for discussion, with PGx associations indicated in parenthesis after the introduction of each drug.

### 3.1. Case 1: Drug–Gene Interactions

The patient was a 65-year-old postmenopausal white woman diagnosed with stage IIB invasive ductal/lobular breast cancer (Appendix A). Immunohistochemistry biomarkers were ER+/PR+/HER2–. The patient received a mastectomy and axilla lymph node dissection, radiation therapy, and adjuvant docetaxel (no CPIC or FDA associations) and cyclophosphamide (*NQO1, SOD2, GSTP1*—CPIC Level D), followed by adjuvant tamoxifen (*CYP2D6*—CPIC Level A, FDA Tier 3). This patient had co-administration of several high-level evidence PGx medications for non-cancer therapy, including ondansetron (*CYP2D6*—CPIC Level A) for nausea, omeprazole (*CYP2C19*—CPIC Level A, FDA Tier 3) for gastric reflux/ulcers, simvastatin (*SLCO1B1*—CPIC Level A, FDA Tier 2) for managing high cholesterol, and hydrocodone (*CYP2D6*—CPIC Level B) for pain management. She was also on medications associated with lower-level evidence PGx, including the opioid oxycodone (*OPRM1, COMT*—CPIC Level C, also relevant to hydrocodone) and the anti-depressant fluoxetine (*CYP2D6, GRIK4*—CPIC Level C). 

Focusing on the high-level PGx evidence, this patient was predicted to be a poor metabolizer (PM) of CYP2D6 based on inheriting two non-functional alleles (*CYP2D6 *4/*5*). Furthermore, she was a rapid metabolizer (RM) for CYP2C19, having inherited one increased function allele (*CYP2C19 *17*). No variants were detected in *SLCO1B1*. The lower-level genes (CPIC Levels C and D) were not evaluated. 

This patient was on four medications that required CYP2D6 for metabolism (tamoxifen, ondansetron, hydrocodone, and fluoxetine). Because tamoxifen and hydrocodone are activated by metabolism, a PM may require higher than standard dosing of those two drugs. Because ondansetron and fluoxetine are inactivated by metabolism, a PM may require lower than standard dosing. Regarding tamoxifen specifically, initial studies showed a direct correlation between CYP2D6 activity, conversion of tamoxifen to its active metabolites endoxifen and 4-hydroxytamoxifen, and outcome [3,21,22]. However, a recent prospective trial of early-stage breast cancer patients receiving tamoxifen found no association between CYP2D6 genotype and outcome [23]. Current guidelines to oncologists from the National Comprehensive Cancer Network (NCCN) do not recommend CYP2D6 testing prior to prescribing tamoxifen, although CPIC evidence predicts a lack of efficacy of tamoxifen in patients with poor CYP2D6 metabolism [24,25]. Furthermore, the NCCN cautions that selective serotonin re-uptake inhibitor (SSRI) anti-depressants such as fluoxetine and paroxetine can lower tamoxifen efficacy. The CPIC did not find sufficient evidence for CYP2D6-guided fluoxetine recommendations. There remains a need for properly designed trials associating the CYP2D6 phenotype to outcome, as long-term survival in ER+ breast cancer is confounded by many factors such as the tumor subtype [1], menopausal status [26], use of aromatase inhibitors [5], and chemotherapy [27]. Since this patient was post-menopausal and was a CYP2D6 PM, an aromatase inhibitor would be preferred over tamoxifen. 

Omeprazole is inactivated by CYP2C19. As this patient was predicted to be a CYP2C19 RM she may have experienced inadequate acid suppression by omeprazole [28,29]. Her medication could have been optimized by either increasing the dose of omeprazole or switching to a proton pump inhibitor that was not metabolized by CYP2C19 [29]. Clinical records were not sufficiently detailed to evaluate the success of these medications or the specific dosing requirements for this patient retrospectively.

### 3.2. Case 2: Drug–Drug Interactions/Phenoconversion

The patient was a 54-year-old perimenopausal white woman diagnosed with stage IIA invasive ductal breast cancer. Immunohistochemistry showed that the tumor biomarkers were ER+/PR–/HER2–. The patient received neoadjuvant chemotherapy with doxorubicin (*SLC28A3*—CPIC Level B/C; *HAS3, NQO1, CBR3*—CPIC Level D), cyclophosphamide (*GSTP1, NQO1, SOD*—CPIC Level D), and paclitaxel (no CPIC or FDA associations), followed by mastectomy and axilla lymph node dissection. She was prescribed capecitabine (*DPYD*—CPIC Level A, FDA Tier 1) after radiation therapy and then given adjuvant tamoxifen (*CYP2D6*—CPIC Level A, FDA Tier 3). High-level PGx evidence is known for tamoxifen, as previously discussed, and capecitabine. Germline variants in *DPYD* are known to predict fluoropyrimidine-associated toxicity from capecitabine or 5-fluorouracil [30], but *DPYD* testing prior to starting these drugs is not standard of care. The patient was also on non-cancer medications including ondansetron (*CYP2D6*—CPIC Level A) for nausea, hydrocodone (*CYP2D6—*CPIC Level B), tramadol (*CYP2D6*—CPIC Level A, FDA Tier 1), and ibuprofen (*CYP2C9*—CPIC Level A) for pain, and paroxetine (*CYP2D6*—CPIC Level A, FDA Tier 3) for depression [31]. She was concurrently prescribed medications associated with lower-level PGx evidence including metoprolol (*CYP2D6*—FDA Tier 3) for high blood pressure.

Focusing on the high-level PGx evidence, this patient was predicted to be a CYP2D6 intermediate metabolizer (IM) with one non-functional allele (*CYP2D6 *4*). No variants were detected in *CYP2C9* or *DPYD*. Although she was predicted to have moderate CYP2D6 activity, the use of multiple CYP2D6 substrates could be detrimental based on competition for enzyme activity, potentially leading to decreased activation of tamoxifen and the opioids, as well as decreased inactivation of ondansetron, paroxetine, and metoprolol [32,33]. Metoprolol plasma concentrations can vary significantly depending on CYP2D6 activity and are associated with bradycardia in PMs [34,35]. Clinical notes for this patient documented that metoprolol was eventually discontinued. Avoiding metoprolol and using a different beta-blocker in this patient could have been beneficial. However, more evidence is needed to support proactive switching of metoprolol or switching to a different anti-depressant based on predicted CYP2D6 phenotype alone. 

### 3.3. Case 3: Non-Actionable but Potentially Relevant PGx Interactions

Patient was a 61-year-old post-menopausal white woman presenting with locally advanced (stage IIIA), triple negative (ER–, PR–, HER2–), invasive ductal breast cancer. She received neo-adjuvant chemotherapy with epirubicin (*CBR3, GSTP1, HAS3, NQO1—*CPIC Level D), cyclophosphamide (*GSTP1, NQO1, SOD—*CPIC Level D), and paclitaxel (no CPIC or FDA associations) followed by mastectomy and radiation therapy. She relapsed approximately 18 months after her initial diagnosis and received additional radiation therapy, as well as adjuvant carboplatin (*MTHF*R—CPIC Level C) and capecitabine (*DPYD—*CPIC Level A, FDA Tier 1). Non-cancer therapy consisted of ondansetron (*CYP2D6—*CPIC Level A) for nausea, as well as morphine (*OPRM1, COMT—*CPIC level C) and tramadol (*CYP2D6—*CPIC Level A, FDA Tier 1; *OPRM1, COMT—*CPIC level C) for pain management. She was also prescribed medical marijuana (no CPIC or FDA associations).

Focusing on the high-level evidence PGx, this patient was a CYP2D6 intermediate metabolizer (IM) with one non-functional allele (*CYP2D6 *4*). No variants were detected in *DPYD*. Similar to case 2, the use of multiple CYP2D6 substrates could decrease enzyme activity, potentially leading to decreased activation of the opioids, as well as decreased inactivation of ondansetron. This patient had a poor overall outcome that could be primarily attributed to her stage of disease and triple-negative biomarker status. Nevertheless, her quality of life may have been improved by optimizing her non-cancer therapy. Of note, the use of recreational (i.e., self-prescribed) and medical marijuana is becoming more common for oncology patients and presents the potential for largely uncharacterized drug–drug and drug–gene associations. Examples of potential PGx interactions with cannabinoids include genes that code for receptors (*CNR1, CNR2, TRPV1, GPR55*), transporters (*ABCB1, ABCG2, SLC6A4*), and drug-metabolizing enzymes (*CYP3A4, CYP2C9, CYP2C19*) [36].

### 3.4. Major Points

Pre-emptive germline PGx testing is not routinely performed in clinical practice in the US in part because of limited support for interpretation and implementation. Drug–drug and drug–gene interactions are very complicated and likely require support for clinical decisions that integrates clinical, demographic, and PGx data. In addition, the FDA does not mandate PGx testing for most drugs. It is clear that the population of patients with breast cancer studied here was exposed to multiple medications, and the existing PGx evidence could have helped guide prescribing decisions. A lack of pre-emptive germline PGx testing may also undermine the evaluation of drug–drug interaction risk because normal/reference phenotypes are assumed. PGx would improve the prediction of drug–drug interaction accuracy and effectiveness during active breast cancer treatment and beyond. Identification of an extreme phenotype could help inform prescribing decisions for biological relatives as well.

Some limitations of our study are as follows: (1) Not all potentially relevant pharmacogenes and variant alleles were included as part of the PGx testing performed; (2) The comprehensive accuracy of pharmacy information in this cohort (e.g., exact timing and dosages of all drugs per patient) was uncertain; and (3) The timing of drug administration (prescription and non-prescription), which requires intensive pharmacy oversight and management, was not evaluated. Prospective implementation goals and content should be designed for future studies, recognizing that drug–gene associations beyond and potentially different from the 38 drug–gene associations described here may have relevant PGx that could benefit other patient populations. 

## 4. Materials and Methods

### 4.1. Subjects

This was a retrospective study of 225 patients with breast cancer treated at the Huntsman Comprehensive Cancer Center (Salt Lake City, UT, USA). A consecutive series of individuals diagnosed with invasive breast cancer (primarily ductal histology) from 2016–2018 was ascertained through Research Informatics Shared Resources databases at the Huntsman Cancer Institute. Patients provided informed consent at the time of diagnosis using IRB-approved protocols (Total Cancer Care—IRB#89989 and Molecular Classifications of Cancer—IRB#10924), which allowed for the collection of biological specimens and clinical data. PGx testing was performed with archived whole blood. Detailed pharmacy and clinical data were obtained for each patient. 

### 4.2. Genetic Analyses and Phenotype Assignment

DNA was extracted from peripheral blood samples using the chemagen M-PVA magnetic Bead Technology and Chemagic MSM I instrument (PerkinElmer Inc., Walthan, MA, USA), and normalized to 50 ng/µL. Samples were evaluated with a targeted OpenArray genotyping panel using TaqMan Real-Time PCR chemistry and commercially available assays on the QuantStudio^TM^ 12K Flex OpenArray^®^ instrument (Thermo Fisher Scientific, Waltham, MA, USA). Data were analyzed using the TaqMan Genotyper software (version 1.3) to assign zygosity for 120 variants in 36 pharmacogenes (see Appendix A). Results for duplicate testing of each sample were further evaluated for agreement using Microsoft Excel. DNA samples diluted to 5 ng/µL were also evaluated for *CYP2D6* copy number using two assays designed to interrogate *CYP2D6* exon 9 (Hs00010001_cn) and intron 6 (Hs04502391_cn) on the QuantStudio^TM^ 12K Flex OpenArray^®^ instrument (Thermo Fisher Scientific, Waltham, MA, USA). Samples were tested in quadruplicate and data were analyzed using the CopyCaller software (version 2.1), with the copy number being assigned [37]. The *1 allele was assigned when none of the targeted variants were detected, for those genes with consensus nomenclature based on “star” alleles. Otherwise alleles were based on the nucleotide detected, compared to the reference nucleotide for the rs number targeted by the assay. All diplotypes and predicted phenotypes were determined based on consensus nomenclature [17,38]. 

### 4.3. Data Analysis

Patients were consented under IRB-approved protocols at the University of Utah to allow researchers to associate the patient’s genetic data with their clinical information. Prescribed drugs and additional clinical information for each patient were extracted from the Huntsman Cancer Center’s Research Informatics Shared Resource (RISR). De-identified samples were provided by the Huntsman Cancer Institute to ARUP Laboratories for analyzing the samples for research (i.e., not for clinical purposes). All PGx data were analyzed using Microsoft Excel and LifeOmic’s (Indianapolis, IN, USA) Precision Health Cloud (PHC). In the PHC, pharmacy and clinical data were aggregated with PGx results as standardized FHIR (http://hl7.org/fhir/) (accessed on 10 January 2021) resource types: Medications, Conditions, Procedures, and Observations. Detailed pharmacy data included medication name, dosage, frequency, and duration. Medications were categorized into drug classes. Within the PHC’s graphical User Interface, visualizations were configured to show longitudinal representation of each patient’s clinical events over time (Appendix A). This clinical timeline view includes medication duration, reason for medication discontinuation, age of diagnosis, date of procedures, and a graphical representation of Karnofsky Performance Status scores over time. In addition to this graphical timeline, all data are also represented in tables. This unified, structured database provides a summary of all indexed data at the patient level, and in aggregate, to support complex querying across data types and patients with the ability to define cohorts and visualize data sets to support this study. No clinical decision support or translational guidance based on PGx data was provided within the PHC product.

## 5. Conclusions

This study illustrates the potential utility of pre-emptive PGx testing in patients with breast cancer. Such information could be used to guide patient therapy and help clinicians make more informed drug treatment decisions to improve patient outcomes and avoid adverse drug events. 

## Figures and Tables

**Table 1 cancers-13-01219-t001:** Drug–gene (germline) pairs with relevance to the patients with breast cancer: anti-cancer therapy.

Drug	Drug Class	Gene	Number of Patients with Drug Noted in Clinical Records (Number of Variant Phenotypes among Those Prescribed the Drug ^1^: %)
Capecitabine	Anti-metabolite	*DPYD*	16 (0%)
Fluorouracil	Anti-metabolite	*DPYD*	12 (0%)
Tamoxifen	Selective estrogen receptor modulator	*CYP2D6*	102 (41 IM, 10 PM, 1 UM: 51%)

^1^ IM: intermediate metabolizer; PM: poor metabolizer; UM: ultra-rapid metabolizer.

**Table 2 cancers-13-01219-t002:** Drug–gene (germline) pairs with relevance to the patients with breast cancer: non-cancer therapy.

Drug	Drug Class	Gene	Number of Patients with Drug Noted in Clinical Records (Number of Variant Phenotypes among Those Prescribed the Drug ^1^: %)
Amitriptyline	Anti-depressant	*CYP2D6* *CYP2C19*	6 (3 IM, 1 PM: 67%) 6 (1 IM, 4 RM: 83%)
Amphetamine	Stimulant	*CYP2D6*	5 (3 IM: 60%)
Aripiprazole	Anti-psychotic	*CYP2D6*	5 (4 IM: 80%)
Aspirin	NSAID ^2^	*G6PD*	61 (PGx not available)
Carvedilol	Anti-hypertensive	*CYP2D6*	15 (5 IM, 2 PM: 47%)
Celecoxib	NSAID	*CYP2C9*	116 (39 IM, 4 PM: 37%)
Ciprofloxacin	Anti-infective	*G6PD*	42 (PGx not available)
Citalopram	Anti-depressant	*CYP2C19*	31 (4 IM, 13 RM, 1 UM: 58%)
Clozapine	Anti-psychotic	*CYP2D6*	1 (1 IM: 100%)
Codeine	Opioid	*CYP2D6*	26 (8 IM, 3 PM, 1 UM: 46%)
Diazepam	Benzodiazepine	*CYP2C19*	33 (3 IM, 10 RM, 3 UM: 50%)
Doxepin	Anti-depressant	*CYP2C19*	5 (1 RM: 20%)
Escitalopram	Anti-depressant	*CYP2C19*	31 (2 IM, 13 RM, 1 UM: 52%)
Esomeprazole	Acid-lowering	*CYP2C19*	7 (2 IM, 1 RM: 43%)
Glipizide	Anti-diabetes	*G6PD*	5 (PGx not available)
Hydrocodone	Opioid	*CYP2D6*	127 (47 IM, 11 PM, 2 UM: 48%)
Ibuprofen	NSAID	*CYP2C9*	144 (44 IM, 5 PM: 34%)
Meloxicam	NSAID	*CYP2C9*	29 (10 IM, 1 PM: 38%)
Metoclopramide	Anti-nausea	*CYP2D6*	19 (8 IM, 1 PM: 47%)
Metoprolol	Anti-hypertensive	*CYP2D6*	21 (9 IM, 2 PM: 52%)
Nitrofurantoin	Anti-infective	*G6PD*	30 (PGx not available)
Omeprazole	Acid-lowering	*CYP2C19*	83 (14 IM, 1 PM, 23 RM, 2 UM: 48%)
Pantoprazole	Acid-lowering	*CYP2C19*	14 (4 IM, 3 RM, 1 PM: 57%)
Paroxetine	Anti-depressant	*CYP2D6*	7 (5 IM: 71%)
Phenazopyridine	Local anesthetic	*G6PD*	12 (PGx not available)
Propranolol	Anti-hypertensive	*CYP2D6*	10 (5 IM, 1 UM: 60%)
Sertraline	Anti-depressant	*CYP2C19*	24 (2 IM, 7 RM, 1 UM: 42%)
Simvastatin	Statin	*SLCO1B1*	10 (7 DF, 1 NF: 80%)
Sulfamethoxazole/Trimethoprim	Anti-infective	*G6PD*	63 (PGx not available)
Tacrolimus	Immunosuppressant	*CYP3A5*	2 (0%)
Tobramycin	Anti-infective	*MT-RNR1*	12 (PGx not available)
Tramadol	Opioid	*CYP2D6*	99 (30 IM, 9 PM, 2 UM: 41%)
Venlafaxine	Anti-depressant	*CYP2D6*	38 (11 IM, 1 PM, 1 UM: 34%)
Voriconazole	Anti-infective	*CYP2C19*	2 (1 RM: 50%)
Warfarin	Anti-coagulant	*CYP2C9* *VKORC1* *CYP4F2*	8 (1 IM, 13%) 8 (4 HET, 1 MUT: 63%) 8 (3 HET: 38%)

^1^ IM: intermediate metabolizer; PM: poor metabolizer; RM: rapid metabolizer; UM: ultra-rapid metabolizer; DF: decreased function; NF: no function; HET: heterozygote for variant allele; MUT: homozygous for variant allele; PGx: pharmacogenetics. “PGx not available” indicates that testing was not performed; however, the frequency of variants is expected to be ~0.2%). ^2^ NSAID: non-steroidal anti-inflammatory drug.

**Table 3 cancers-13-01219-t003:** Number of breast cancer patients (%) predicted to exhibit variant metabolic phenotypes for common pharmacogenes involved in drug metabolism.

Gene	Poor Metabolizer (PM)	Intermediate Metabolizer (IM)	Normal Metabolizer (NM)	Rapid Metabolizer (RM)	Ultra-Rapid Metabolizer (UM)
*CYP2C9* (*n* = 219)	7 (3%)	72 (33%)	140 (64%)	None	None
*CYP2C19* (*n* = 218)	4 (2%)	41 (19%)	104 (47%)	58 (26%)	11 (5%)
*CYP2D6* (*n* = 220)	20 (9%)	81 (37%)	115 (52%)	None	4 (2%)

## Data Availability

The datasets used and/or analyzed during the current study are available from the corresponding author on reasonable request.

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
