# Peer review of "Potential Utility of Pre-Emptive Germline Pharmacogenetics in Breast Cancer"

_cancers, 2021, doi:10.3390/cancers13061219_

Round 1

Reviewer 1 Report

This topic is of great important to oncologists in treating their breast cancer patients who will all require polypharmacy.

Major points

  • Consider reorganizing any discussion of pharmacotherapy (e.g., anti-cancer vs. non-cancer) and genes (e.g., germline vs. somatic) into groups.
  • Strongly recommend that a medical oncologist be included on this manuscript. The presentation of the cases is not in the usual way an oncologist would present and it is unclear regarding the types of therapy the patients received. Also, trastuzumab is not a TKI but a monoclonal antibody (page 3, line 108).
  • More detail is needed regarding the use of these PGx data. Were they available for clinical care? If so, were you able to assess whether the results changed prescribing?
  • The comments on tamoxifen appear to be overstated based on more recently available and more rigorous evidence (e.g., PMID: 30676859). CPIC guidelines have not been updated since more rigorous evidence has become available. Furthermore, NCCN guidelines do not support CYP2D6-guided tamoxifen therapy as this is a highly controversial area. Two cases should be chosen that emphasize the supportive care therapy and how PGx results affect use and dosing of these rather than the primary endocrine therapy where the efficacy of changing dose or drug is unclear.
  • In the discussion, consider commenting further on information to support clinicians as they attempt to incorporate PGx into prescribing decisions. I can envision readers being overwhelmed by the high levels of clinically actionable results without being supported in learning where to learn more about how to act on such results.

Abstract: Abstract is too focused on background and laboratory methodology and does not have enough information about results.

Introduction:

  • Gene names should be itaclicized
  • “Pharmacy decisions” is an odd phrase. Perhaps the authors meant prescribing decisions?
  • Use full abbreviations for CYP enzmes (e.g., CYP2D6 rather than 2D6)
  • This statement seems misguided as it is not based on the newest data that includes clinical outcomes: “Patients that are CYP2D6 poor metabolizers may 65 not respond well to tamoxifen therapy[18,19].”
  • Paragraph two ends awkwardly with the sentence “Genes associated with pharmacodynamics…” How are they relevant? For which drugs? The sentence seems like an after thought in an attempt to comment on both PK and PD.
  • I am not sure it is accurate to say that the FDA has published “levels of evidence associated with PGx drug-gene pairs.”
  • Re: “198 drug-gene pairs,” perhaps it makes sense to describe drug-gene pairs with clinically relevant levels of evidence (e.g., A, B) since that was just defined and is more useful information
  • Paragraph 3 does a good job at explaining how the genes were selected, maybe in this paragraph also mention how they narrow down to 38 drug-gene associations
  • Perhaps it is more accurate to state “patients with breast cancer” rather than “breast cancer patients.”

Results:

  • Why is are the following listed as a drug-gene pair with relevance to breast cancer patients?
    • amphetamine-CYP2D6; proporanolol-CYP2D6
  • 2 Pharmacy Trend Analysis section- Recommend explicitly mentioning drugs such as ibuprofen, hydrocodone, celecoxib, tamoxifen, and tramadol which has a high number of patients with variant phenotype
  • 3 PGx Genotyping panel and CYP2D6 copy number variation analysis- Paragraph 1 do you have data on whether prescribing actions
  • Table 2- Include the (N=#) for each gene
  • Table 3- Row 4 SLCO1B1 be consistent and mention what actions need to be taken given predicted phenotype

Discussion:

  • Are the cases actual patient data from the cohort? If so, why were these two cases selected?
  • General comments on cases
    • Germline and somatic data are discussed together in a confusing way.
    • It seems odd to mention genes (e.g., OPRM1) with lower levels of evidence for a clinical case as their lower level of evidence would preclude them from being a clinical consideration.
  • Case 1
    • 184 mentions patient being on ondansetron, what effect would patient being a CYP2D6 poor metabolizer have on this medication?
    • Where these changes mentioned in end of paragraph 2 such as increasing the dose of PPI or patient being at high risk for dose related toxicity from fluoxetine reported to the patient and/or provider? Were these changed made?
    • Again, the comments on tamoxifen appear to be overstated based on more recently available and more rigorous evidence (e.g., PMID: 30676859). Furthermore, NCCN guidelines do not support CYP2D6-guided tamoxifen therapy.
  • Consider expanding a little bit more on how perspective studies should be designed

Materials and Methods:

  • 1 Subjects- Please include how long these patients were followed
  • 1 Subjects- were there any exclusion criteria?
  • 2 Genetic analyses and phenotype assignment- Line 272 “All diplotypes and predicted phenotypes were” is missing the end of the sentence
  • 3 Data analysis- Recommend including if duration of therapy for each medication identified was taken into consideration; were all prescriptions regardless of duration included (ie. oxycodone to treat acute pain for 3 days vs. bactrim for UTI vs. blood pressure medications taken chronically)
  • A description of what LifeOmic software is or how it works would be beneficial to the reader. Is it acting similar to PharmGKB and providing CPIC levels and FDA information?

Author Response

Please see the "bolded" responses to the reviewer in the attachment

Reviewer 2 Report

In the present paper by Bernard et al, authors focus their attention on the importance of pre-emptive pharmacogenetic (PGx) testing, in this case for breast cancer patients, to improve drugs efficacy and, thus, patient outcome, minimizing adverse events (AEs). Although PGx is a very bedated subject and there are a lot of publications available on this topic, I believe that it is never redundant to reiterate its clinical importance, even in retrospective studies like the present one.

The paper is well written, clear and comprehensible. However, I recommend to accept it for publication after minor revisions. 

Specifically, the criticisms concern the number of discussed cases, i.e 2 cases on a total of 225 breast cancer patients, and the structure of the case discussion. Is it possible, for the authors, to add other relevant cases to the discussion section? Furthermore, for the selected cases, it is possible to compare the obtained PGx results with their real clinical outcome (as was done for case 2 [line 221-223] but in more detail) in order to further stress the utility of pre-emptive PGx testing in clinical practice?

Author Response

Thank-you for the encouraging comments and constructive suggestions.  As this was a retrospective study, there are very limited “real” clinical outcomes to include.  The providers and patients have not been provided the PGx data so even speculative impact is not possible.  Throughout the revised manuscript it is emphasized that this is a retrospective study for which clinical actions based on PGx were not possible.  The case descriptions were, however, expanded, and a medical oncologist that specializes in breast cancer management was added to the list of authors.

Reviewer 3 Report

Summary:

The pharmacogenomics is one of the steps to the personalized medicine. The authors purpose a study with 225 female breast cancer patients to understand the potential utility of pre-emptive germline pharmacogenetics in breast cancer. Although, for now, there are some points the authors need to improve, this is a very interesting theme. The pharmacogenomics is one of the steps to the personalized medicine.

Broad comments:

When it comes to breast cancer, all variables could influence the treatment and consequently, the patient results. This study becomes relevant by crossing the variables that could influence the response to drugs. It presents potential relations that could be interesting to explore in some practical cases. It reveals a path to the personalized medicine.

Regarding weaknesses, the authors could more clearly show the results of the crossing of variables and its impact on the patients’ treatment and outcomes. The discussion should be more detailed. And the conclusion more complete. There are several questions that the authors talk about at the abstract and the introduction, but they are not referred at the conclusion.

Specific comments:

Line 12 – Before the phrase about germline variation, explain (with a short but sustained phrase what is a germile);

Line 52 – Before “Pharmacogenetics refers to germline variations…” It is useful to define what is pharmacogenetics and pharmacogenomics, and the differences between those concepts;

Line 54/55 – Explain why pharmacokinetics and pharmacodynamics are the two major processes responsible for drug response and adverse events. The authors could use a definition of those processes;

Line 59 – Why the BRCA mutation is used? The authors could write something about it;

Line 65 – The authors refer poor metabolizers, but they do not say nothing about the other kinds of metabolizers and don’t refer the importance of this. The authors should explain better those concepts and its importance, as they will use it many times in at the results;

Line 95 – “We propose that the inclusion of PGx information in the medical management of cancer patients… “When? At what stage of the planning?... Could the authors explain this better;

Line 105 to 107 – Author could place this information in a more, visually, appealing way (maybe a table);

Line 134 – The authors should explain here or before why they consider that the most problematic gene is CYP2D6;

Line 157 – The authors should explain the reference to warfarin in this context;

Discussion:

Facts: 225 breast cancer patients and 38 drug-gene associations. Why the authors only chose two patient case examples to discuss? The authors should justify their option.

Conclusion:

At line 95 the authors wrotewe propose that the inclusion of PGx information in the medical management of cancer patients…”. They could explore it better at the conclusions.

Round 2

Reviewer 1 Report

This paper has been much improved with the author’s careful and considerable edits. However, there are still some outstanding issues that should be addressed.

  1. Was the data de-identified since it is stated to be retrospective. It is unclear as to how patients would have PGx laboratory results obtained, was that based on Total Care consent? Does this cover also obtaining genetic data, is the GINA information in the consent? Was a separate waiver of consent obtained since this is identified genetic data?
  2. “Conflicting studies have led to a discrepancy between National Comprehensive Cancer Network guidelines and CPIC guidelines in terms of recommending CYP2D6 genotyping prior to prescribing tamoxifen” (page 12). This is not an accurate statement as CPIC does not make recommendations to test. This is the problem, NCCN does not recommend testing which is the basic guideline that oncologists use for making practice decisions. From NCCN V1.2021: “Some SSRIs like fluoxetine and paroxetine decrease the formation of endoxifen, 4-OH tamoxifen, and active metabolites of tamoxifen, and may impact its efficacy. Caution is advised about coadministration of these drugs with tamoxifen. However, SNRIs (citalopram and venlafaxine) appear to have minimal impact on tamoxifen metabolism. At this time, based on current data the panel recommends against CYP2D6 gene testing for patients being considered for tamoxifen. Coadministration of strong inhibitors of CYP2D6 should be used with caution.”
  3. I still do not like the emphasis on tamoxifen for both the patient cases since this is not a recommendation to test by oncology clinical guidelines as stated above and has been hotly debated. Also, no data supports a dose change or less efficacy with a lower dose in fact the opposite is true. (DeCensi, Randomized Placebo Controlled Trial of Low-Dose Tamoxifen to Prevent Local and Contralateral Recurrence in Breast Intraepithelial Neoplasia, J Clin Oncol 2019;37:1629-37). I think this detracts from the advantages to using pre-emptive PGx testing for supportive care or polypharmacy in these cases. I would not cloud the points with that issue. It will confuse rather than enlighten oncologists and discourage rather than encourage use of pre-emptive PGx.
  4. Case 1 has a few additional issues:
    1. It is confounded by fluoxetine being a strong CYP2D6 inhibitor per the FDA Table of Substrates, Inhibitors and Inducers (see Table 3-2). Genotype would not be helpful here as the patient is a CYP2D6 poor metabolizer while prescribed fluoxetine. Suggest removing fluoxetine from this case. Another reason to remove fluoxetine is the lack of CPIC guideline recommendations for fluoxetine as they did not find sufficient evidence to support CYP2D6-guided fluoxetine recommendations
    2. Case 1 also misses an opportunity to tie in CPIC guideline recommendations and instead defers to comments like: “theoretically require lower than standard dosing.” It would be more helpful to relay clear recommendations, like CPIC guideline recommendations, for drugs selected in a case. If CPIC does not have a recommendation to change therapy for that phenotype, as some of the drug-gene pairs in this case, then I would suggest reconsidering the drug-gene pairs selected.

Minor comments

  • Page 5: It would be helpful to define CPIC level A or B in the manuscript for those readers who are not familiar.
  • Throughout the manuscript suggest listing CPIC levels instead of stating “high-level evidence PGx.” or “lower-level evidence.” It is more descriptive, consensus-based, and the authors already introduced what these levels mean. For example
    • Original manuscript: high PGx evidence including ondansetron (CYP2D6) for nausea; hydrocodone (CYP2D6), tramadol (CYP2D6) 777 and ibuprofen (CYP2C9) for pain; and paroxetine (CYP2D6) for depression
    • Proposed change: clinically relevant PGx data including ondansetron (CYP2D6 – CPIC Level A) for nausea; hydrocodone (CYP2D6– CPIC Level B), tramadol (CYP2D6 – CPIC Level A) 777 and ibuprofen (CYP2C9 – CPIC Level A) for pain; and paroxetine (CYP2D6 – CPIC Level A) for depression

Author Response

  1. Was the data de-identified since it is stated to be retrospective. It is unclear as to how patients would have PGx laboratory results obtained, was that based on Total Care consent? Does this cover also obtaining genetic data, is the GINA information in the consent? Was a separate waiver of consent obtained since this is identified genetic data? We inserted the following in lines 309-314: Patients were consented under IRB-approved protocols at the University of Utah to allow researchers to associate the patient’s genetic data with their clinical information. Prescribed drugs and additional clinical information for each patient were extracted from the Huntsman Cancer Center’s Research Informatics Shared Resource (RISR). De-identified samples were provided by the Huntsman Cancer Institute to ARUP Laboratories Inc for analyzing the samples for research (i.e. not for clinical purposes).
  2. “Conflicting studies have led to a discrepancy between National Comprehensive Cancer Network guidelines and CPIC guidelines in terms of recommending CYP2D6 genotyping prior to prescribing tamoxifen” (page 12). This is not an accurate statement as CPIC does not make recommendations to test. This is the problem, NCCN does not recommend testing which is the basic guideline that oncologists use for making practice decisions. From NCCN V1.2021: “Some SSRIs like fluoxetine and paroxetine decrease the formation of endoxifen, 4-OH tamoxifen, and active metabolites of tamoxifen, and may impact its efficacy. Caution is advised about coadministration of these drugs with tamoxifen. However, SNRIs (citalopram and venlafaxine) appear to have minimal impact on tamoxifen metabolism. At this time, based on current data the panel recommends against CYP2D6 gene testing for patients being considered for tamoxifen. Coadministration of strong inhibitors of CYP2D6 should be used with caution.” We inserted the following sentences (lines 204-209) to more accurately reflect differences in the guidelines: Current guidelines to oncologists from the National Comprehensive Cancer Network (NCCN) do not recommend CYP2D6 testing prior to prescribing tamoxifen, although CPIC evidence predicts a lack of tamoxifen efficacy in patients with poor CYP2D6 metabolism[24,25]. Furthermore, the NCCN cautions that SSRI’s, like fluoxetine and paroxetine, can also lower tamoxifen efficacy. CPIC did not find sufficient evidence for CYP2D6-guided fluoxetine recommendations. I still do not like the emphasis on tamoxifen for both the patient cases since this is not a recommendation to test by oncology clinical guidelines as stated above and has been hotly debated. Also, no data supports a dose change or less efficacy with a lower dose in fact the opposite is true. (DeCensi, Randomized Placebo Controlled Trial of Low-Dose Tamoxifen to Prevent Local and Contralateral Recurrence in Breast Intraepithelial Neoplasia, J Clin Oncol 2019;37:1629-37). I think this detracts from the advantages to using pre-emptive PGx testing for supportive care or polypharmacy in these cases. I would not cloud the points with that issue. It will confuse rather than enlighten oncologists and discourage rather than encourage use of pre-emptive PGx. We have substantially de-emphasized tamoxifen in case 2.
  1. Case 1 has a few additional issues:
    1. It is confounded by fluoxetine being a strong CYP2D6 inhibitor per the FDA Table of Substrates, Inhibitors and Inducers (see Table 3-2). Genotype would not be helpful here as the patient is a CYP2D6 poor metabolizer while prescribed fluoxetine. Suggest removing fluoxetine from this case. Another reason to remove fluoxetine is the lack of CPIC guideline recommendations for fluoxetine as they did not find sufficient evidence to support CYP2D6-guided fluoxetine recommendations. Changed the discussion in lines 213-216.
    2. Case 1 also misses an opportunity to tie in CPIC guideline recommendations and instead defers to comments like: “theoretically require lower than standard dosing.” It would be more helpful to relay clear recommendations, like CPIC guideline recommendations, for drugs selected in a case. If CPIC does not have a recommendation to change therapy for that phenotype, as some of the drug-gene pairs in this case, then I would suggest reconsidering the drug-gene pairs selected. We have restricted recommendations to CPIC and NCCN guidelines.

 Minor comments

  • Page 5: It would be helpful to define CPIC level A or B in the manuscript for those readers who are not familiar. We have added the CPIC definitions of evidence levels when first mentioned (line 72)
  • Throughout the manuscript suggest listing CPIC levels instead of stating “high-level evidence PGx.” or “lower-level evidence.” It is more descriptive, consensus-based, and the authors already introduced what these levels mean. For example
    • Original manuscript: high PGx evidence including ondansetron (CYP2D6) for nausea; hydrocodone (CYP2D6), tramadol (CYP2D6) 777 and ibuprofen (CYP2C9) for pain; and paroxetine (CYP2D6) for depression.
    • Proposed change: clinically relevant PGx data including ondansetron (CYP2D6 – CPIC Level A) for nausea; hydrocodone (CYP2D6– CPIC Level B), tramadol (CYP2D6 – CPIC Level A) 777 and ibuprofen (CYP2C9 – CPIC Level A) for pain; and paroxetine (CYP2D6 – CPIC Level A) for depression. This is a great suggestion and has been implemented in the revision; thank-you.